# Recent Strategies for Cancer Therapy: Polymer Nanoparticles Carrying Medicinally Important Phytochemicals and Their Cellular Targets

**DOI:** 10.3390/pharmaceutics15112566

**Published:** 2023-11-01

**Authors:** Metin Yıldırım, Melike Sessevmez, Samet Poyraz, Nejat Düzgüneş

**Affiliations:** 1Department of Biochemistry, Faculty of Pharmacy, Harran University, Sanliurfa 63050, Turkey; metinyildirim4@gmail.com; 2Department of Pharmaceutical Technology, Faculty of Pharmacy, Istanbul University, Istanbul 34116, Turkey; melikesessevmez@gmail.com; 3Department of Analytical Chemistry, Faculty of Pharmacy, Harran University, Sanliurfa 63050, Turkey; s.poyraz88@gmail.com; 4Department of Biomedical Sciences, Arthur A. Dugoni School of Dentistry, University of the Pacific, San Francisco, CA 94103, USA

**Keywords:** polymeric nanoparticles, cancer, phytochemicals, antioxidants, flavonoids

## Abstract

Cancer is a leading cause of death in the world today. In addition to the side effects of the chemotherapeutic drugs used to treat cancer, the development of resistance to the drugs renders the existing drugs ineffective. Therefore, there is an urgent need to develop novel anticancer agents. Medicinally important phytochemicals such as curcumin, naringenin, quercetin, epigallocatechin gallate, thymoquinone, kaempferol, resveratrol, genistein, and apigenin have some drawbacks, including low solubility in water, stability and bioavailability issues, despite having significant anticancer effects. Encapsulation of these natural compounds into polymer nanoparticles (NPs) is a novel technology that could overcome these constraints. In comparison to the free compounds, phytochemicals loaded into nanoparticles have greater activity and bioavailability against many cancer types. In this review, we describe the preparation and characterization of natural phytochemical-loaded polymer NP formulations with significant antioxidant and anti-inflammatory effects, their in vitro and in vivo anticancer activities, as well as their possible cellular targets.

## 1. Introduction

Cancer remains the most common cause of death worldwide. According to the World Health Organization, there were close to 10 million deaths due to cancer in 2020 (https://www.who.int/news-room/fact-sheets/detail/cancer (accessed on 7 September 2023)). Chemotherapy, radiotherapy, hormone therapy, surgery, and immunotherapy are commonly used to treat cancer. Conventional cancer treatments have several challenges, including serious side effects, a high risk of recurrences, drug resistance, and limited therapeutic efficacy [1]. To overcome these limitations, new anticancer drugs with enhanced efficacy and fewer side effects are required.

The term “phytochemical” was first coined in the 1950s, but the medicinal effects of phytochemicals have been known for centuries [2]. Phytochemicals are naturally occurring substances found in all plant components, including the leaves, stems, flowers, and roots, and are also responsible for the colors, flavors, and aromas of fruits, vegetables, grains, nuts, and seeds. Many phytochemicals have anticancer, antioxidant, anti-inflammatory, and disease-preventing properties.

Phytochemicals can be divided into two main categories. Primary metabolites are crucial to the plant’s survival. Secondary metabolites are non-essential to the plant’s growth and development and include terpenes, carotenoids, and flavonoids [3,4,5]. Curcumin, naringenin, quercetin, epigallocatechin gallate, resveratrol, thymoquinone, genistein, and apigenin have been utilized extensively in preclinical and clinical cancer treatment trials. The poor water solubility, low bioavailability, and rapid metabolism of phytochemicals have prompted the exploration of new therapeutic strategies for their use in cancer therapeutics. One of these strategies is the use of nanoparticles to deliver therapeutic phytochemicals [6].

Nanoparticles are a cutting-edge technology that is altering the way scientists and engineers approach many different fields of study. Polymer nanoparticles, in particular, are enabling the development of novel drug delivery, material engineering, and diagnostic solutions [7]. In this review, we describe some of the most successful applications of nanoparticles in delivering phytochemicals to particular cancer cells in vitro and in vivo.

## 2. Nanoparticle Preparation and Characterization

The selection of the nanoparticle formulation technique depends on the physicochemical properties of the drug and the desired characteristics of the NPs. Solvent evaporation, ionic gelation, and nanoprecipitation are three methods commonly used for the preparation of NPs and nanocarriers. Each method has its advantages and limitations, and the selection of the appropriate method should be based on factors such as drug solubility, drug stability, and the scalability of the method.

Emulsion-solvent evaporation is widely used and easy to perform. This method involves the dissolution of the polymer and the drug or active compound in an organic solvent and emulsifying the mixture into an aqueous phase, followed by the evaporation of the solvent to form NPs. Emulsion-solvent evaporation allows for control over the NP size and drug loading, but it may not be suitable for hydrophilic drugs. For this reason, the double emulsion-solvent evaporation method is used, but it is not easy to scale up for final NP production [8].

Ionic gelation is a mild method for the preparation of polymeric NPs based on the electrostatic interaction of oppositely charged polyelectrolytes in aqueous solution. This method is relatively simple and does not require specialized equipment; it is particularly suitable for hydrophilic drugs and can be used to achieve sustained drug release. However, it may not be suitable for hydrophobic drugs because of their poor solubility in water [9,10].

The nanoprecipitation method is relatively simple but may require the use of organic solvents; it involves the rapid mixing of a polymer and a drug solution in a non-solvent or a surfactant solution. This leads to the formation of NPs through the precipitation of the polymer–drug complex. This method has shown benefits in terms of stability, reproducibility, time efficiency, and scale-up; however it may not be suitable for hydrophilic molecules [9].

The size range of polymer NPs used for antitumor drug delivery is typically 50 nm to 300–500 nm [11]. These particles have unique, designable properties such as size, shape, surface charge, and drug delivery capabilities, making them adaptable for particular applications [12]. The choice of NP material and the size of nanoparticles can impact their effectiveness in cancer therapy. The size of NPs can influence their biodistribution, cellular uptake, and therapeutic efficacy [13]. Smaller NPs have been shown to exhibit enhanced cellular uptake and penetration into tumor tissues, leading to improved drug delivery and therapeutic outcomes [13].

The use of polymeric particles may increase the solubility of poorly water-soluble antitumor phytochemicals, increase their bioavailability, decrease the necessary dose of the drug, and reduce the frequency of administration, thereby potentially reducing the toxicity and side effects of the drug. NPs may also enable the targeted delivery of the phytochemicals to the tumor through both active targeting via their engineered affinity to receptors overexpressed on tumor cells, and also via passive targeting, i.e., through the enhanced permeability and retention (EPR) effect. Furthermore, polymeric NPs can be designed to respond to stimuli including reduced pH, redox potential, magnetic fields, and ultrasound for drug release in tumors [14,15].

The physicochemical characteristics of nanoparticles have been investigated by techniques including dynamic light scattering (DLS), transmission electron microscopy (TEM), scanning electron microscopy (SEM), and atomic force microscopy (AFM). DLS is a technique widely used to measure the size distribution of nanoparticles in a liquid suspension. By analyzing the fluctuations in the intensity of scattered light, DLS can provide information about the hydrodynamic diameter of nanoparticles and their stability in solution [16]. TEM allows for high-resolution imaging of nanoparticles, providing information about their size, shape, and structure [17]. SEM may be used for nanoparticle characterization and provides detailed images of the surface of nanoparticles. AFM can provide high-resolution images of nanoparticles, enabling the analysis of their size, shape, surface topography, and mechanical properties [16].

Polymer nanoparticles can be loaded with an active therapeutic agent and used in drug delivery applications. The particle size and surface charge can be adjusted to deliver the drug to the desired location in the body, resulting in greater efficacy and fewer side effects. Polymer nanoparticles are also used in diagnostic applications to detect and measure biomarkers, including proteins and DNA, in blood or other body fluids, allowing diseases such as cancer to be diagnosed and therapeutic treatments to be monitored [18,19,20]. Several polymeric nanoparticles have been used for encapsulating natural (Figure 1) and synthetic anticancer agents. These include dendrimers [21], poly(lactic-co-glycolic-acid) (PLGA) [22,23], poly(lactide) (PLA) [24], PHEMA [25], chitosan [26], and polymeric micelles [27].

Below, we describe in detail the anticancer effects of the nanoparticle formulations of curcumin, naringenin, quercetin, epigallocatechin gallate, resveratrol, thymoquinone, genistein, and apigenin. Table 1 summarizes the various cellular effects and signaling pathways of phytochemical-loaded nanoformulations in different types of cancers.

## 3. Curcumin

Curcumin (Cur) is a polyphenol found in ground rhizomes of *Curcuma longa* L., a plant native to South Asia. It is a powerful antioxidant that has been used in traditional medicine for centuries to treat a variety of ailments, including inflammation, pain, digestive issues, and skin conditions. It is not soluble in water at acidic and neutral pH [64,65]. Cur has been investigated recently for its potential to treat a variety of illnesses, including cancer [66], diabetes, cardiovascular disease, Alzheimer’s disease, and depression [67]. Cur not only has remarkable anti-inflammatory and antioxidant properties [23] but also significant anti-cancer activity. It interacts with several crucial cellular pathways, including PI3K/Akt, JAK/STAT, MAPK, Wnt/β-catenin, p53, NF-κB, and apoptosis-related signaling pathways [68]. Cur downregulates the expression and activity of the epidermal growth factor receptor (EGFR) and the human epidermal growth factor receptor 2 (HER2) [69]. It interferes with the cell cycle at the G2/M phase [70] and regulates cancer-related miRNA expression [68].

Cur-loaded polymeric nanoparticles could prevent the growth of cancer cells by enhancing its effectiveness and water solubility [28]. Cur-loaded PLGA nanoparticles (Cur-PLGA) were prepared using the solvent evaporation method to overcome solubility problems and to enhance the intracellular delivery of Cur in triple-negative breast cancer. This formulation increased Cur solubility by an order of magnitude and enhanced its cytotoxicity by a factor of 3. Moreover, the expression of the transcriptional regulators, i.e., HIF-1α and nuclear p65 (RelA), associated with hypoxia and most types of cancer was reduced significantly with Cur-PLGA treatment of MDA-MB-231 breast cancer cells and A549 lung cancer cells [28].

A single-emulsion method was used to optimize the formation of uniform Cur-bearing PLGA nanoparticles (Cur-PLGA NPs) for improved anticancer treatment against breast cancer [29]. DLS analysis revealed that the mean particle size and zeta potential of the formulation were 120 nm and −30 mV, respectively. SEM results showed that Cur-PLGA NPs were spherical in shape with diameters of 116.9 ± 3.8 nm. Cur-PLGA NPs inhibited the proliferation of MCF-7 human breast cancer cells by blocking the G2/M phase of the cell cycle [29].

To overcome the multidrug resistance (MDR) problem of PTX against breast cancer, Lin et al. [30] fabricated paclitaxel (PTX)- and Cur-loaded PEG-PLGA NPs (PC-NPs). PC-NPs had a spherical shape, small size (85.8 nm), good encapsulation efficiency (83.05% for PTX and 85.84% for Cur) and drug loading (38.7% for PTX and 32.22% for Cur), a smooth-surface morphology, and a suitable negative charge (−20.3 mV). PC-NPs significantly suppressed the NF-κB signaling pathway and inhibited the production of P-glycoprotein involved in drug efflux and hence drug resistance. Moreover, the viability of MDA-MB-231 cells was reduced more efficiently compared to PTX treatment alone [30].

Cholesterol-conjugated PLA micelles encapsulating Cur (Cur-mPEG-PLA-Ch) were designed to treat MDA-MB-231 human breast cancer and B16F10 murine melanoma cells. Cur-mPEG-PLA-Ch had higher cytotoxicity than free Cur in both cell lines as well as in murine-xenografted tumors. The uptake of Cur in the nanoparticles by both cell types was several-fold higher than the free drug [31]. Although Cur-encapsulating poly(amidoamine) (PAMAM) dendrimers inhibited the proliferation of and telomerase activity in T47D breast cancer cells, they did not cause cytotoxicity [32].

The viability of A549 human alveolar cell carcinoma and Hela human cervical carcinoma cells was reduced significantly by biodegradable Ferrite@FA-CUR-PLA-MMS treatment, with IC_50_ values of 10.2 µg/mL and 8.8 µg/mL, respectively [33]. These biodegradable Cur-loaded magnetic microspheres were prepared by using the dual-emulsion (water-in-oil, oil-in-water) solvent washout method that produced smooth-surfaced, spherical microspheres [33].

There are very few clinical studies of Cur nanoparticles. The poor bioavailability of curcumin prompted Kanai et al. [71] to utilize a nanoparticle formulation (Theracurmin), which consisted of curcumin (10%), other curcuminoids (2%), glycerin (46%), gum ghatti (4%), and water (38%) and had a mean diameter of 190 nm. Theracurmin at a dose of 150 mg administered orally to healthy volunteers in capsules resulted in curcumin plasma levels of 189 ± 48 ng/mL, which compared favorably with curcumin levels obtained in other studies after a dose of 8 g. At a dose of 210 mg Theracurmin, a curcumin level of 275 ± 67 nm/mL could be obtained.

A nanoparticle formulation of curcumin, SinaCurcumin, 9.5 nm in diameter and composed of micelles prepared with GRAS (generally recognized as safe) excipients, was administered to patients with localized, muscle-invasive bladder cancer undergoing induction chemotherapy at a dose of 180 mg/day [72]. In a previous study, this nanoformulation was found to increase the bioavailability of curcumin by about 59-fold compared to the free phytochemical [73]. This formulation was tolerated well by the patients, and there was a complete clinical response in 50% of the patients, although this response was not statistically significant compared to the placebo group (31%).

## 4. Naringenin

Citrus fruits, including grapefruit, oranges, and tangerines, are rich in naringenin (Nar), a compound that has the potential to treat a number of diseases, including diabetes, obesity, and cancer [74]. Thanks to its antioxidant and anti-inflammatory properties, Nar can help prevent cell damage caused by free radicals while also raising the levels of hormones that assist in controlling metabolism [75]. Nar may assist people with type 2 diabetes lower their blood sugar levels as well as their cholesterol levels and risk factors for cardiovascular disease [76]. It may also help protect against cancer by inducing apoptosis or programmed cell death, as observed with pancreatic cancer cells [77]. Nar triggers an increase in the levels of caspase 3, caspase 8, Caspase 9, and BAX expression, all of which are associated with inducing apoptosis in cancer cells [78].

Nar can induce programmed cell death in cancerous cells, including breast, liver, colon, and prostate [79,80,81,82]. By modulating various signaling pathways, it can trigger intrinsic or extrinsic apoptotic pathways, effectively leading to cell death in tumors. This flavonoid can intervene in the progression of the cell cycle, arresting cells in the G2/M phase [83]. By hindering the cell cycle, Nar can prevent the proliferation of cancer cells. Chronic inflammation is often associated with cancer progression. Nar possesses anti-inflammatory properties, potentially reducing the risk or progression of inflammation-associated cancers [84]. It acts as a potent antioxidant, neutralizing free radicals. This action prevents oxidative DNA damage, which is a precursor to mutagenesis and carcinogenesis. The spread of cancerous cells to other parts of the body, known as metastasis, is a significant concern in cancer progression. Nar can inhibit certain enzymes and signaling pathways associated with cell invasion and migration, thereby potentially preventing metastasis. To grow, tumors need to develop new blood vessels in a process called angiogenesis. Nar has shown the potential to inhibit factors promoting angiogenesis, thereby limiting tumor growth. Various cellular signaling pathways can be dysregulated in cancer [85]. Nar can modulate several of these pathways, including the PI3K/AKT and MAPK pathways, affecting the growth and survival of cancer cells [82].

Nar-loaded PLGA nanoparticles prepared by the emulsion-diffusion evaporation method were used to treat pancreatic cancer cells in vitro. The nanoparticles had a uniform spherical shape, a diameter of about 150 nm, a negative surface charge, high stability, and better cytotoxic activity against pancreatic cancer cells compared to free Nar [34]. To improve Nar bioactivity against A549 lung cancer cells, Kumar et al. developed Nar-loaded chitosan NPs (Nar-Chi-NPs). Nar-Chi-NPs, which were prepared with a 5:1 (*w*/*w*) ratio of chitosan to sodium tripolyphosphate, had an average size of 407 nm and increased the cytotoxicity of Nar against A549 cells by about 2-fold compared to the free compound at various concentrations. Indicating their potential biosafety, they were not toxic to normal 3T3 fibroblasts [35].

In another study, pH- and thermo-sensitive PHEMA-NPs incorporating Nar were formulated to enhance cytotoxic activity against MCF-7 human breast cancer cells (Figure 2) [36]. The maximum release of Nar from NPs occurred at 41 °C, pH 6.0. Nar-SNPs significantly decreased the viability of the cells compared to free Nar. The NPs induced early apoptosis and increased the percentage of cells in the G1 phase of the cell cycle in a dose-dependent manner [36].

The anti-cancer activity of Nar-NPs was evaluated in vitro in the A549 cell line and in vivo in the urethane-induced lung carcinoma model in Wistar albino rats [37]. Polycaprolactone (PCL) NPs containing Nar (Nar-HA@CH-PCL-NPs) were prepared using the nanoprecipitation method to enhance the water solubility and oral bioavailability of Nar [37]. The surface of the NPs was then decorated with hyaluronic acid to target them to the CD44 receptor on lung cancer cells. The targeted Nar-HA@CH-PCL-NPs were taken up by A549 cells to an extent 1.5 times higher than Nar-PCL-NPs without the hyaluronic acid. The Nar-HA@CH-PCL-NP treatment induced cell cycle arrest in the G2/M phase in A549 cells and, following delivery by the oral route, reduced tumor development of lung cancer in the animal model [37].

Wang et al. developed pH-sensitive and aptamer-modified Nar-loaded polymer micelles (AP-Nar@ZIF-8 PMs) for use in treating MCF-7 and A549 human cancer cells [38]. AP-Nar@ZIF-8 PMs have a spherical shape and uniform dispersity, with a diameter of 239.8 nm and a zeta-potential of −8.9 mV. In 48 h, 91.4% of Nar was released from AP-Nar@ZIF-8 PMs at pH 5. The viability of A549 cells was inhibited significantly by AP-Nar@ZIF-8 PMs compared to free Nar [38].

To the best of our knowledge, there are no reports of clinical trials involving Nar nanoparticles.

## 5. Quercetin

Found in many fruits, vegetables, and herbs, including apples, onions, and green tea, quercetin (QCT) is a naturally occurring flavonoid that has been researched for its potential to support the health of many organ systems [86]. QCT, which may be taken as a supplement in capsule or powder form, has also been evaluated for its ability to reduce the risk of cancer [87], diabetes [88], cardiovascular disease [89], and the symptoms of allergies, asthma [90], and other respiratory conditions. As a powerful antioxidant, QCT protects the body from oxidative stress by reducing inflammation and neutralizing free radicals [91]. QCT induces apoptosis in a variety of cancer cells (e.g., MDA-MB-231, Hep G2, HT-29, and HER2-expressing breast cancer cells) by activating intrinsic and extrinsic pathways, which are two main signaling pathways of apoptosis. The extrinsic pathway is activated by signals from outside the cell, while the intrinsic pathway is activated by internal stress signals. Both pathways ultimately lead to the activation of caspases, a family of proteases that initiate a cascade of proteolytic events that dismantle and remove the dying cell. QCT regulates the activation of Caspase 3, Caspase 8, Caspase 9, and p53 phosphorylation, and it inhibits STAT3 signaling, the PI3K/AKT survival pathway, and the MAPK and NF-κB pathways, which are crucial in cancer development and progression [92].

In one study, QCT-loaded, pH-sensitive polymer nanoparticles (Eudragit® S100 (QCT-NPs) were designed to achieve colon-specific drug delivery at neutral pH and to avoid release in the low pH of the stomach. They were prepared in deionized water, where the polymer is expected to be in its acidic form, using the nanoprecipitation method. The nanoparticles had a mean diameter of 66.8 nm and a slightly negative surface charge of −5.2 mV [39]. QCT release began when the pH was changed to 7.2 and reached 91.8% after 24 h. QCT-NPs showed 80-fold higher cytotoxicity than free QCT against CT26 murine colon carcinoma cells [39]. Chitosan nanoparticles containing QCT (QCT-Chi NPs) were formulated with 79.8% encapsulation efficiency. QCT-Chi NPs significantly decreased the viability of human lung cancer cell (A549) and human breast cancer cells (MDA MB 468) [40].

The anticancer activity of QCT-loaded PLGA nanoparticles (PLGA-QCT NPs), formulated by the solvent evaporation method, was investigated in human cervical (HeLa) and breast cancer cell lines (MCF-7). Apoptosis, mitochondrial damage, caspase activation, and cell cycle arrest were induced by treatment with PLGA-QCT NPs, which also caused the upregulation of FoxO1 and the downregulation of PI3K and p-Akt; these effects caused a decrease in the viability of the cells [41]. In another study, QCT-loaded PLGA/TPGS (d-α-tocopherol polyethylene glycol 1000 succinate) nanoparticles were prepared by using a single-step nanoprecipitation method, producing particles with a uniform spherical morphology with a mean diameter of about 198 nm and good drug loading capacity (8.1%) [42]. QCT-PLGA/TPGS NPs inhibited the proliferation of MDA-MB 231 human breast cancer cells more effectively than free QCT and decreased the migration of the cells in a wound-healing experiment as well as the invasion ability of the cells in a Transwell assay. The migration inhibition was attributed to the inhibition of urokinase-type plasminogen activator, which promotes tumor metastasis. These NPs were also effective in inhibiting tumor growth and metastasis in an orthotopic murine mammary carcinoma model [42].

Application of QCT-caffeic-acid phenethyl ester (CAPE)-co-loaded PLGA nanoparticles (QCT-Ca NPs) upregulated the expression of the main proteins of the intrinsic apoptosis pathway, caspase 3 and caspase 9, in HT-29 colon cancer cells [43]. These nanoparticles were prepared using the single emulsion (*o/w*) solvent evaporation method [43]. The mean diameter of the QCT-Ca NPs was about 238 nm, with a polydispersity index of 0.34. The encapsulation efficiencies of QCT and CAPE were 74% and 65%, respectively.

Ren et al. [44] prepared nanoparticles with gold-QCT and poly (DL-lactide-co-glycolide) and evaluated their activity against the liver cancer cell lines MHCC97H, Hep3B, HCCLM3, and Bel7402. These particles induced apoptosis by enhancing the activity of Cyto-c/caspase signaling and also suppressed liver cancer cell growth through Akt/ERK1/2, AP-2β/hTERT, and p65/COX-2 signaling inactivation, albeit at relatively high concentrations, possibly resulting from the dense colloidal gold particles [44]. The QCT-NPs injected intraperitoneally into mice bearing MHCC97H xenografts also caused a significant reduction in tumor volume [44].

The potential anticancer activity of QCT nanoparticles have not been investigated in clinical trials.

## 6. Epigallocatechin Gallate

Epigallocatechin gallate (EGCG) is the most abundant catechin in green tea (*Camellia sinensis*) and makes up 50% of the total catechins found in beverages made from the tea leaves [93]. EGCG has strong antioxidant and anti-inflammatory properties, which may help protect against certain types of cancer, cardiovascular disease, and other illnesses [94,95]. EGCC has been shown to suppress the growth of tumor cells and the production of new blood vessels within tumors, preventing cancer spread [96]. Cancers of the bladder, brain, breast, and liver might be susceptible to its effects. As an antioxidant, EGCG not only neutralizes harmful free radicals but also induces apoptosis, suppresses cell proliferation, and hampers tumor growth. EGCG can affect various cell signaling pathways essential for cell growth, survival, and differentiation via its effects on the ERK, MAPK, PI3K/AKT, and JAK/STAT pathways. It can inhibit the activity and expression of growth factors like EGF, which often plays a role in the progression of various cancers [94,97]. Increased insulin sensitivity and reduced risk of type 2 diabetes have been shown in studies investigating EGCG’s role in promoting metabolic health [98]. Additionally, it may help reduce cholesterol levels and improve cardiovascular health [99].

To treat prostate cancer cells in spheroids, Alserihi et al. [45] functionalized EGCG-loaded NPs with folic acid to mediate binding to both the human folate receptor alpha (FOLR1) and prostate-specific membrane antigen (PSMA). EGCG-loaded NPs had a regular spherical shape, and the particle size ranged from 115 nm to 130 nm. PC3 cells and 22Rv1 spheroids were used to evaluate the efficiency of targeted therapy. The anti-proliferative activity of EGCG against prostate cancer cells was increased significantly following treatment with the targeted NPs, particularly in the case of the PSMA^+^ cells that express higher levels of FOLR1. Furthermore, mitochondrial depolarization decreased (15%), and polarization (18%) and apoptotic cells (12%) increased following treatment of 22Rv1 spheroids with EGCG-targeted NPs (Figure 3) [45].

In the lung cancer cell lines A549 and H1299, EGCG-NP treatment improved by 3–4 fold the anti-proliferative activity compared to free EGCG by inducing apoptosis. In this study, the PLGA NPs loaded with EGCG were formulated by the oil-in-water emulsion solvent evaporation method and had a mean diameter of about 176 nm [46]. EGCG-NPs blocked NF-κB activity and suppressed genes regulated by NF-κB, such as *bcl*-2, *bcl-xL*, *cox*-2, and *c-myc*, and the genes for TNF-α, cyclinD1, TWIST, and MMP2 [46].

EGCG-loaded chitosan NPs (Chi-EGCG) were targeted to the folate-binding protein (FBP) by conjugating folic acid to the NPs [47]. The presence of folic acid increased the particle size of Chi-EGCG NPs from about 164 nm to 342 nm. Chi-EGCG NPs were stable and biocompatible and had anti-proliferative activity in MCF-7 breast cancer cells but at relatively high concentrations (0.15–0.3 mM) [47].

EGCT-NPs have not been explored in clinical trials relating to cancer therapy.

## 7. Thymoquinone

Thymoquinone (TQ) is a natural compound found in the seeds of the *Nigella sativa* plant, also known as black cumin. It has been used in traditional medicine for centuries to treat a variety of ailments, including digestive issues and skin conditions. Recent studies have revealed that TQ has potential medicinal uses, with anti-inflammatory, anti-cancer, and antioxidant properties [100,101]. TQ may be beneficial in decreasing the risk of cardiovascular diseases such as hypertension and atherosclerosis by reducing the inflammation associated with arthritis and asthma [102,103]. TQ has also been investigated as a potential anti-cancer agent to inhibit the growth of prostate, breast, and lung cancer cells [104]. TQ inhibits tumor angiogenesis and the NF-κB, Akt, and extracellular signal-regulated kinase signaling pathways, which are involved in tumor cell survival and proliferation, respectively [105].

The cytotoxic effect of TQ on the colon cancer cell line C26 and the co-delivery of free doxorubicin (DOX) in different TQ NP formulations was evaluated with the MTT assay [48]. TQ was incorporated into lipid polymer NPs (LPNPs) composed of phosphatidylcholine and PLGA, using single-emulsion solvent evaporation [48]. The wound-healing assay was also utilized to assess cell migration. At a simulated intestinal fluid pH of 6.8, LPNPs containing TQ had an average diameter of 184 nm and a loading efficiency of 60%, and they mediated more drug release than polymeric NPs. PLGA-PC-TQ NPs improved DOX’s anticancer activity more than TQ NPs. The uptake of PLGA-PC-TQ by Caco-2 cells was 2.5 times greater than that of NPs without phosphatidylcholine. This study demonstrated that the anticancer activity of DOX can be improved with the combination of PLGA-PC-encapsulated TQ and free DOX (Figure 4) [48].

To preserve the biological activity of TQ before reaching the target sites, Noor et al. [49] encapsulated TQ in PLGA-PEG and Pluronic F68 (PF68) nanoparticles prepared using an emulsion-solvent evaporation technique. They then evaluated the cytotoxic activities of the nanoparticles against tamoxifen-resistant (TamR) MCF-7 breast cancer cells (Figure 5). The size of TQ-PLGA-PF68 nanoparticles was 77 × 27 nm, and the encapsulation efficiency was 94%. A higher concentration of TQ-PLGA-PF68 nanoparticles was needed to reach the IC_50_ for toxicity to (TamR) MCF-7 cells compared to the parental MCF-7 cells. The two other resistant subtypes, the PTX-resistant (PacR) MDA-MB 231 triple-negative breast cell line and TamR UACC732 inflammatory breast cancer cells, needed lower concentrations of TQ-PLGA-PF68 nanoparticles than their respective parental cell lines. Based on these findings, TQ encapsulation with PLGA-PEG and Pluronic F68 appears to be a promising anti-cancer agent that can overcome resistance to chemotherapy in breast cancer [49].

To enhance the anticancer effects of TQ, Alshehri et al. [50] produced TQ-encapsulating chitosan-(Chi)-PLGA nanoparticles (TQ-CS-PLGA-NP) using the emulsion evaporation method. The TQ-release rate was significant and prolonged in TQ-PLGA-NPs and TQ-CS-PLGA-NPs. In comparison to uncoated TQ-PLGA-NPs and TQ suspension, optimized TQ-CS-PLGA-NPs had significantly greater muco-adhesion, intestinal permeability, enhanced antioxidant potential, and higher cytotoxicity against MDA-MB-231 and MCF-7 cells [50].

Transferrin (TF), a biodegradable, non-immunogenic, non-toxic, iron-transporting protein can be used as a targeting ligand to enhance the delivery of TQ into non-small cell lung carcinoma (NSCLC) cells, a lethal cancer with few treatment options. PEGylated PLGA TQ nanoparticles (TQ-NP) with attached TF enhanced the activity of TQ in A549 cells [51]. TF-TQ-NPs were shown to involve the p53 and ROS feedback loop in regulating microRNA (miRNA) circuitry to control apoptosis and migration of NSCLC cells. The simultaneous activation of miR-34a and miR-16 targeting BCL-2 to induce apoptosis in A549 cells was promoted by TF-TQ-NP-mediated p53 upregulation. In addition, TF-TQ-NPs restricted cell migration via actin de-polymerization by activation of the p53/miR-34a axis. In vivo tests with Balb/c mice confirmed the anticancer activity of these nanoparticles against NSCLC [51].

To target TQ to ovarian cancer cells, radio-iodinated folic acid–chitosan-conjugated TQ nanoparticles (FATQChi-NP) were formulated [52]. Cell viability was determined in SKOV-3 and Caco-2 cells treated for 48 h. While the IC_50_ values for TQ and TQCS were about 12.6 µg/mL, the IC_50_ for FATQCS was about 7.6 µg/mL, indicating the advantage of targeting the folic acid receptor on the SKOV-3 cells. By contrast, folic acid conjugation did not confer an advantage in the treatment of Caco-2 cells. The uptake of TQ in the targeted NPs was about twice as much in SKOV-3 cells as that in Caco-2 cells, providing an explanation for the higher cytotoxicity in SKOV-3 cells [52]. TQ nanoparticles were also effective against A375 human melanoma cells; Ibrahim et al. [53] produced PLGA NPs incorporating TQ with a high efficiency (97%). While the IC_50_ of TQ alone was 50 µg/mL, that of TQ-PLGA NPs was about 5 µg/mL. Importantly, the authors also reported the instability of free TQ in the medium.

We are not aware of any clinical trials involving TQ nanoparticles.

## 8. Kaempferol

Kaempferol (KPF), a flavanol found in a variety of plants and foods such as grapes, onions, apples, oranges, tea, and beans, is a powerful antioxidant with numerous potential pharmacological effects, including the reduction of inflammation and the improvement of heart health [106]. The ability to reduce inflammation is the most powerful of KPF’s diverse pharmacological properties, including antioxidant, antimicrobial, anti-inflammatory, antiviral, and anti-cancer effects and cardiovascular protection [107]. KPF has been shown in animal tests to lessen inflammation and the risk of chronic illnesses like heart disease and diabetes [108,109,110]. KPF also reduces the risks of certain cancers such as prostate cancer. KPF inhibits the growth and spread of cancer cells by inhibiting the androgen-dependent pathway and suppressing vasculogenic mimicry and invasion [111]. KPF has been found to modulate the estrogen receptor alpha (ERα) signaling pathway and to inhibit the growth of MCF-7 breast cancer cells through various mechanisms, including apoptosis and cell cycle arrest [112]. In ovarian cancer cells, KPF is able to effectively induce apoptosis by activating p53 in the intrinsic pathway while simultaneously suppressing the process of angiogenesis [113].

Luo et al. [54] developed five different NP formulations encapsulating KPF: poly(ethylene oxide)-poly(propylene oxide)-poly(PEO-PPO-PEO), PLGA, chitosan, polyethyleneimine (PEI), and PAMAM. The NPs were prepared by the nanoprecipitation method and were evaluated for their effectiveness in inhibiting the viability of cancerous and normal ovarian cells. Among the nonionic polymeric nanoparticles, KPF-(PEO-PPO-PEO) caused a significant reduction in the viability of both cancerous and normal cells. KPF-PLGA nanoparticles provided a remarkable reduction in cancer cell viability compared to KPF alone, without causing a significant reduction in the viability of normal cells. Both PEO-PPO-PEO and PLGA nanoparticle formulations of KPF outperformed KPF alone in terms of cytotoxicity to cancer cells [54].

No clinical trials have been carried out on the utility of KPF nanoparticles.

## 9. Resveratrol

Resveratrol (RSV), a natural stilbene and a non-flavonoid polyphenol known for its antioxidant properties and found in red grape skin, red wine, and some herbs, is gaining interest as a natural health supplement with potential benefits for a variety of health disorders [114]. RSV has antibacterial, antioxidant, anticancer, and antiaging effects [115]. Perhaps most importantly, RSV can reverse multidrug resistance in cancer cells, and when combined with conventional chemotherapeutic drugs, it can enhance the activity of the drugs against cancer cells [116]. RSV’s anti-tumor activity is mediated by its ability to suppress hypoxia-inducible factor-1α, vascular endothelial growth factor, and Src as well as induce apoptosis by activating p53 [55]. It can influence numerous cell signaling pathways, including those involved in cell survival, growth, and differentiation, such as the mitochondrial intrinsic apoptotic pathway, Wnt, PI3K/AKT, and MAPK pathways [22,116].

In terms of its cardiovascular advantages, resveratrol reduces the risk of heart disease and stroke by enhancing the function of the artery lining and the heart’s ability to pump blood [117]. Not only does RSV reduce the risk of cancer by limiting the production of chemicals that cause cancer, but it also helps prevent the formation of amyloid plaques in the brain that are associated with Alzheimer’s disease [116].

To investigate the in vitro and in vivo activity of RSV against CT26 colon cancer cells, Jung et al. [55] prepared RSV-loaded polyethylene glycol-polylactic acid (RSV-PEG5k–PLA5k) polymer nanoparticles (NPs). Treatment with RSV-NPs at doses of 40 µM and 20 µM for 72 h resulted in the reduction of the number of cancer cells in culture to 5.6% compared to untreated controls, and their ability to form colonies was decreased to 6.3%. Flow cytometry and Western blots revealed increased apoptotic cell death, while RSV-NPs significantly reduced ^18^F-fluorodeoxyglucose uptake and the formation of reactive oxygen species. In the in vivo studies, the administration of RSV-NPs intravenously to CT26 tumor-bearing mice resulted in a reduction in ^18^F-fluorodeoxyglucose uptake. In comparison to controls that had blank NPs injected, treatment with RSV-NPs led to retardation of tumor growth and an improvement in survival [55]. Nassir et al. [56] investigated the cytotoxic and apoptotic cell death pattern of RSV-PLGA nanoparticles against the prostate cancer cell line LNCaP. The NPs were characterized by TEM, capture efficiency, DSC, and drug release. RSV-PLGA NPs inhibited LNCaP cell growth by 50% and 90% at 15.6 and 41.1 µM, respectively. At all tested concentrations, RSV-PLGA NPs exhibited significantly greater cytotoxicity to LNCaP cells than free RSV. The findings provide credence to the possibility that RSV-loaded NPs could be used in the treatment of prostate cancer without causing any adverse effects as long as they are localized sufficiently in the prostate [56].

Sanna et al. prepared RSV-loaded Poly(epsilon-caprolactone) (PCL) and PLGA-PEG conjugate (PCL: PLGA-PEG-COOH) NPs using the nanoprecipitation method and evaluated them for the controlled delivery of bioactive RSV for prostate cancer chemoprevention and chemotherapy [57]. Cellular uptake of NPs was measured using confocal fluorescence microscopy in the prostate cancer cell lines DU-145, PC-3, and LNCaP, and antiproliferative activity was evaluated using the MTT assay. RSV was loaded successfully onto PCL: PLGA-PEG-COOH NPs, producing NPs with an average diameter of 150 nm and encapsulation efficiencies ranging from 74% to 98%. Fifty-five percent of RSV was released within 7 h when NPs were exposed to pH 6.5 and 7.4, imitating the acidic tumor microenvironment and physiological conditions, respectively. The nano-RSV significantly enhanced cytotoxicity compared to free RSV in the concentration range 10 µM to 40 µM for all three cell lines [57].

Vijayakumar et al. [58] formulated RSV-loaded PLGA-D-alpha-tocopheryl poly(ethylene glycol) 1000 succinate blend nanoparticles (RSV-PLGA-BNPs) by the single-emulsion solvent-evaporation technique to evaluate their chemotherapeutic activity against glioma in vivo. After intravenous injection, Charles Foster rats underwent pharmacokinetic and tissue distribution analysis. RSV-PLGA-BNPs were internalized by C6 glioma cells and caused increased cytotoxicity compared to free RSV. A hemocompatibility investigation revealed that intravenous injection of RSV-PLGA-BNPs is safe. RSV-PLGA-BNPs remained in circulation for up to 36 h, which is roughly 18 times longer than the half-life of RSV solution in plasma. The higher accumulation of RSV-PLGA-BNPs in the brain compared to free RSV suggested that RSVPLGA-BNPs could be a promising agent for increasing systemic circulation and plasma half-life while also having superior anticancer activity against glioma [58]. In an attempt to increase RSV plasma stability and to decrease its rate of metabolism, Yee et al. [59] conjugated RSV to a low-molecular-weight copolymer, methoxypoly(ethylene glycol)-PLA (mPEG-PLA), and formulated the conjugated RSV, together with free RSV, to produce RSV-mPEG-PLA NPs. The C57BL/6J mouse model with subcutaneous B16-F10 melanoma tumors was used to investigate the anticancer efficacy of these formulations. Compared to nanoparticles encapsulating free RSV in mPEG-PLA (encapsulated RSV NPs) and free RSV alone, conjugated RSV NPs exhibited a consistent plasma stability profile and a decrease in liver metabolic rate. Conjugated RSV NPs showed a much greater tendency to decrease the tumor volume compared to free RSV and encapsulated RSV-NPs. Thus, chemical conjugation of RSV with NPs has the potential to be further developed for the inhibition of early tumor growth in vivo [59].

RSV-NPs have not been investigated in clinical trials, according to the *PubMed* database.

## 10. Genistein

Genistein (Gen), a phytoestrogen present in many legumes, such as soybeans, beans, and chickpeas, is used extensively as a dietary supplement due to its possible health advantages, which range from the treatment of osteoporosis to the prevention of cancer [118,119]. Recent studies have suggested that Gen has activity against different types of cancer cells, including breast cancer. Gen binds to estrogen receptors in the body and blocks the action of estrogen, which can reduce the risk of estrogen-related cancers. Gen has also been found to inhibit the growth of tumor cells and stop the spread of cancer cells [120]. Gen induces programmed cell death in cancer cells through the intrinsic and extrinsic apoptotic pathways [121]. It can inhibit the progression of the cell cycle, particularly at the G2/M phase, thus impeding the proliferation of cancer cells [122]. Tyrosine kinases play significant roles in cell signaling related to growth and differentiation. Gen can inhibit tyrosine kinases, thereby disrupting several pathways crucial for cancer cell survival and growth [123]. It can also neutralize free radicals, thereby preventing oxidative damage to DNA that would otherwise lead to cancer. Gen affects various cellular signaling pathways, including the ERK1/2, PI3K/AKT, and MAPK pathways that are vital for the growth, differentiation, and survival of cancer cells [124].

In addition to its potential cancer-fighting effects, Gen has also been found to have positive effects on bone health. Gen has been associated with increased bone mineral density, reduced bone loss, and the risk of osteoporosis as well as improved cholesterol levels and reduced inflammation [60,125].

Gen- and temozolomide (TMZ)-loaded PLGA NPs (Gen-TMZ-NP) were developed to combat glioblastoma multiforme [61]. The encapsulation efficiency was determined in single and dual drug-loaded NPs; the encapsulation efficiency of dual-drug-loaded NPs was slightly lower than both single-drug-loaded NPs. The particle size of Gen-TMZ-NPs was in the range 136–181 nm. SEM of Gen-TMZ-NPs showed a spherical shape and uniform dispersity of the particles. Treatment of U87MG cells with Gen-TMZ-NPs potently inhibited the viability of the cells and increased the levels of Cyt-c, Caspase-3 Caspase-9, and BAX. These results indicate that Gen-TMZ-NPs triggered the intrinsic apoptosis pathway in these cells [61].

To treat ovarian cancer, Patra et al. [62] fabricated folate receptor-targeted, Gen-loaded PLGA-PEG NPs (Gen-PLGA-PEG-FA NPs) by the nanoprecipitation method. Folate-decorated NPs bound to folate receptors and were internalized by ovarian cancer cells overexpressing the folate receptor. Folic acid decoration slightly increased the particle size of Gen-PLGA-PEG NPs (104 to 125 nm). The uptake of Gen-PLGA-PEG-FA NPs was higher than that of Gen-PLGA-PEG NPs in SKOV-3 cells, and they exhibited superior anticancer activity compared to the non-targeted NPs [61]. In another study, Rahmani et al. [126] prepared chitosan NPs encapsulating Gen for the treatment of human colorectal cancer cells (HT-29). The particle size was larger than that of the NPs described above (788 nm). Gen-C-NPs had a spherical morphology with a smooth surface. The encapsulation efficiency was approximately 77% (*w*/*w*). Gen-C-NPs showed cytotoxic activity against HT-29 cells, and they were not toxic to normal human epithelial cells [126].

Like most of the nanoformulations of phytochemicals discussed above, except curcumin, Gen-NPs have not been investigated in clinical trials.

## 11. Apigenin

Apigenin (APG), a natural flavonoid found in high amounts in some herbs, including chamomile, parsley, celery, and thyme, as well as some fruits, such as oranges, lemons, and limes, has antioxidant and anti-inflammatory effects [127]. The anti-cancer activity of APG is very limited due to its poor water solubility (1.35 µg/mL). APG increases glutathione, catalase, and SOD expression and results in the inhibition of metastasis and angiogenesis by affecting signaling molecules in the mitogen-activated protein kinase (MAPK) pathway. The signaling molecules in this pathway include the c-Jun N-terminal kinases (JNK), the extracellular-signal-regulated kinase (ERK), and the p38 MAPK signaling pathway [127].

To treat melanoma lung metastasis, APG-loaded PLGA NPs were fabricated by a nanoprecipitation method and functionalized with meso-2,3-dimercaptosuccinic acid (DMSA) [63]. DMSA-conjugated APG-NPs decreased the proliferation of B16F10 (melanoma) and A549 (lung carcinoma) cells around 3.5 and 7.8 times more effectively compared to free APG, respectively. DMSA-conjugated APG-NPs increased mitochondrial membrane potential, thus leading to apoptosis, and accumulated in lung tissue [63]. Perhaps most significantly, DMSA-conjugated APG-NPs virtually eliminated B16F10 tumors in the lungs (Figure 6) [63].

As far as we are aware, APG-NPs have not been investigated clinically.

## 12. Conclusions

In this review, we examined the anticancer efficacy of natural phytochemical compounds when they are incorporated into different polymer NPs because of the limitations of the compounds on their own, such as low bioavailability, solubility in water, and stability. Phytochemical compounds were loaded into natural, synthetic, biodegradable, and non-biodegradable polymer nanoparticles such as PLA, PGA, Eudragit, PLGA, PAMAM, chitosan, and gelatin, and their anticancer effects were evaluated. The prepared nanoformulations were characterized by methods such as IR, SEM, DLS, and TEM. The in vivo and in vitro anticancer effects of the described phytochemical-loaded NPs were assessed against breast, prostate, ovarian, lung, colon, and pancreatic cancer cells. Overall, in in vivo and in vitro experiments, phytochemical compound-loaded polymer NPs suppressed tumor development after numerous doses without noticeable side effects or tumor necrosis. By contrast, NP-free phytochemical compounds were ineffective against tumor growth and caused fluid seepage or inflammation. However, one of the potential limitations of polymeric NPs is the amount of phytochemicals they can incorporate [128]. These findings provide the background and motivation for the future development and implementation of nanotechnology-based chemotherapeutic techniques for improved cancer therapy.

## Figures and Tables

**Figure 1 pharmaceutics-15-02566-f001:**
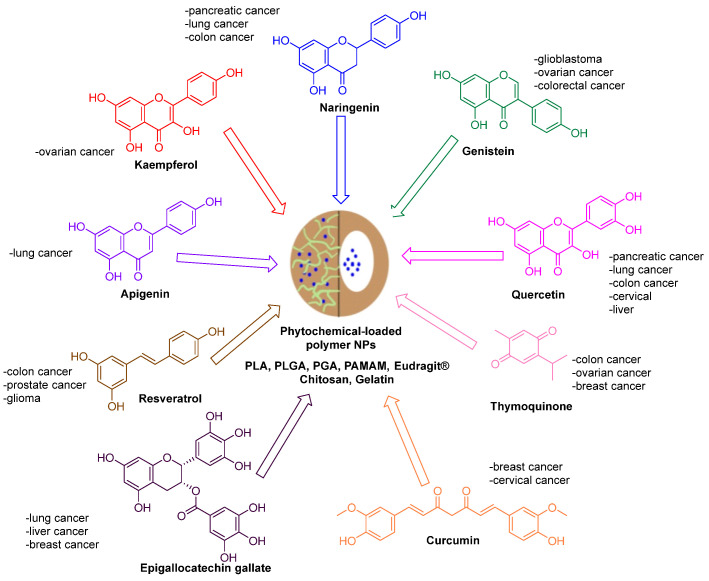
Phytochemical compounds loaded in polymeric NPs and their anticancer activities.

**Figure 2 pharmaceutics-15-02566-f002:**
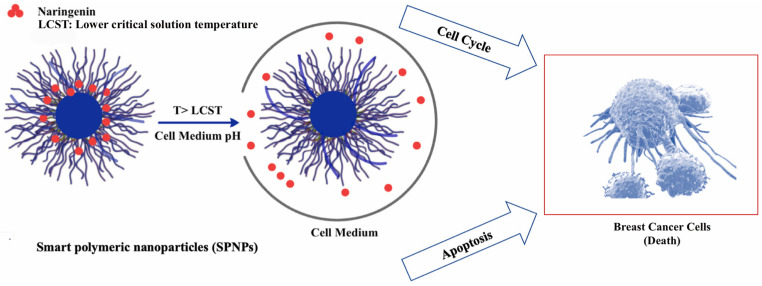
Naringenin-loaded, pH- and thermo-sensitive smart polymeric nanoparticles showed cytotoxic activity against human breast cancer cells (reproduced with permission from Yıldırım et al., 2022 [36]).

**Figure 3 pharmaceutics-15-02566-f003:**
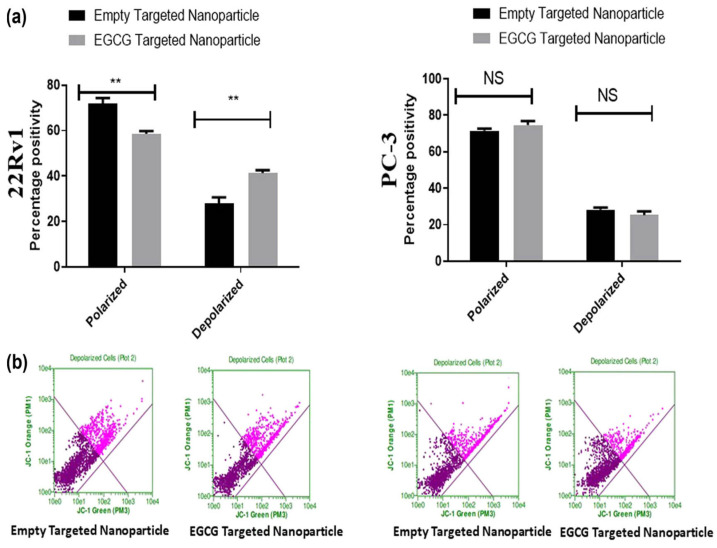
Epigallocatechin gallate (EGCG) encapsulated in targeted nanoparticles induces depolarization in spheroid cultures. Both 22Rv1 (**a**) and PC3 (**b**) cells were grown on poly-HEMA-coated plates and treated with either empty targeted nanoparticles or EGCG in targeted nanoparticles (6 μM) for 6 days. At the end of the time-point, JC-1 dye was added to the treated spheroids and incubated for 30 min at 37 °C. The fluorescence intensity was measured using a Guava Flow Cytometer at the standardized wavelength provided by the manufacturer. Values are shown as mean ± SEM (*n* = 6). ** *p* < 0.01; NS: no significance (reproduced with permission from Alserihi et al., 2022 [45]).

**Figure 4 pharmaceutics-15-02566-f004:**
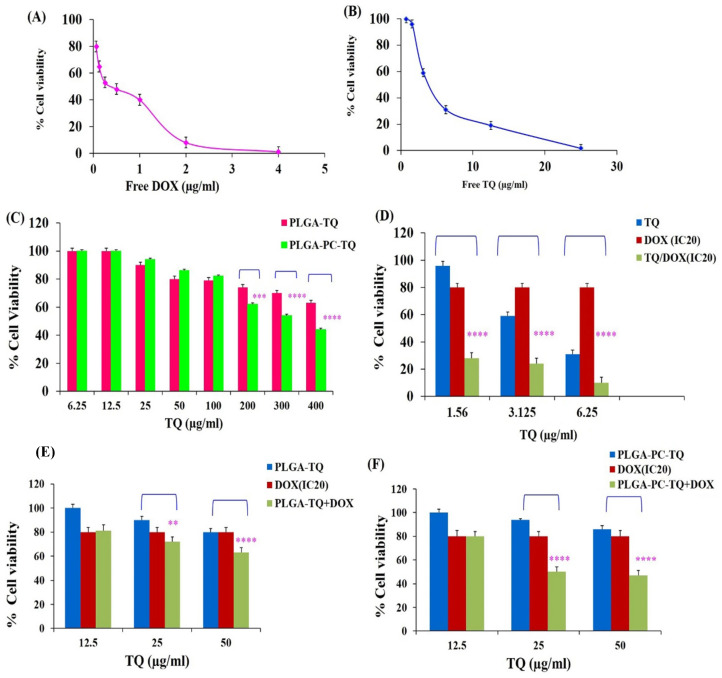
MTT cell viability assay of the effect of different formulations of thymoquinone (TQ) and their co-delivery with free doxorubicin (DOX) on the colon cancer cell line C26. Cell viability following treatment with (**A**) free DOX, (**B**) free TQ, and (**C**) NPs at various concentrations of DOX and TQ (*n* = 3). Cell viability of DOX (IC20), which was co-delivered with (**D**) free TQ, (**E**) PLGA/TQ, and (**F**) PLGA-PC/TQ in C26 cells, measured by the MTT assay (*n* = 3). * indicates a significant difference between two types of NPs. The asterisks indicate the following *p*-values: ** *p* ≤ 0.01, *** *p* ≤ 0.001 and *****p* < 0.0001 (reproduced with permission from Moghaddam et al., 2021 [48]).

**Figure 5 pharmaceutics-15-02566-f005:**
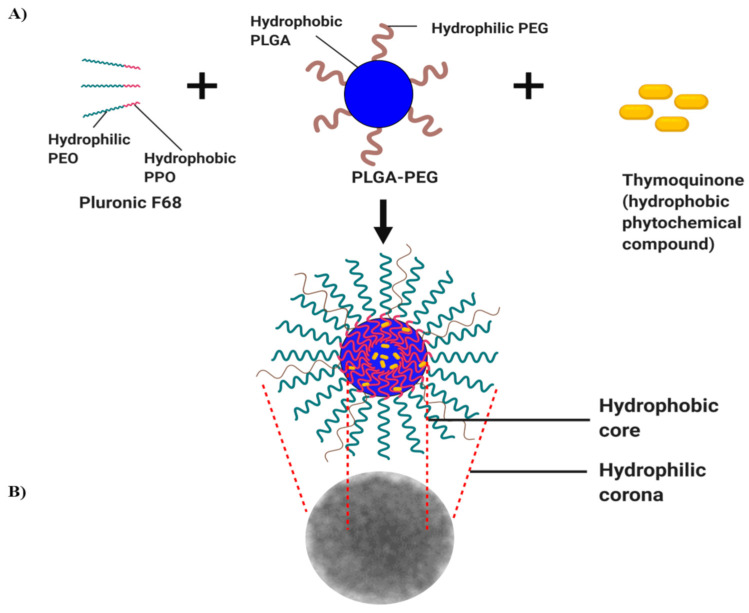
Illustration of the preparation of thymoquinone (TQ) nanoparticles. (**A**) TQ self-assembles with an amphiphilic molecule of Pluronic F68 and PLGA-PEG, forming a hydrophobic core, while intact TQ is located within the core. (**B**) TEM observation of TQ-PLGA-PF68 nanoparticles (reproduced from Noor et al., 2021 [49]).

**Figure 6 pharmaceutics-15-02566-f006:**
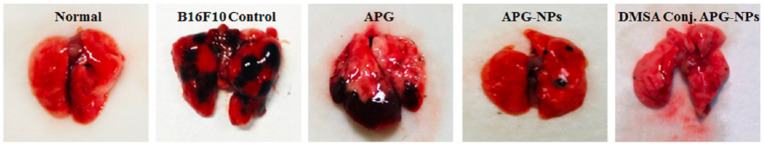
Tumor colonies representing the in vivo anti-pulmonary metastasis (B16F10) efficacy of different formulations of apigenin (APG). Four- to six-week-old C57BL/6 mice (20–25 g body weight) were divided randomly into six groups (n = 6), and melanoma lung metastasis was induced by injecting B16F10 cells (1 × 10^6^ in 100 μL of PBS) intravenously (via the tail vein). Metastatic nodule formation was reduced in mice treated with free APG (5 mg/kg body weight) and APG-NPs (dose equivalent to conjugated NPs). The lungs of mice treated with the DMSA-conjugated formulation had much lower tumor mass and number compared to the other treatment groups and the saline-treated controls (reproduced with permission from Sen et al. [63]).

**Table 1 pharmaceutics-15-02566-t001:** Various cellular effects and signaling pathways of phytochemicals loaded nanoformulations in different types of cancers.

	Drug Carrier	Cancer	Cellular Effect	Cell Line	Particle Size	Method	References
Curcumin	PLGA	Breast	Decreases HIF-1α and Nuclear p65 activity	MDA-MB-231	606 nm	Solvent evaporation	[28]
Curcumin	PLGA	Breast	G2/M block of cell cycle progression	MCF-7	116.9 ± 3.8	Single emulsion	[29]
Curcumin	PLGA	Breast	Suppresses the NF-κB signaling pathway and inhibits the production of P-glycoprotein	MDA-MB-231	85.8 ± 0.21 nm	Antisolvent precipitation	[30]
Curcumin	PLA	Breast	Inhibits cell proliferation	T47D	169.3 ± 1.52 nm	Thin-film hydration method	[31]
Curcumin	PAMAM dendrimers	Breast	Increases telomerase activity	T47D	NI *	Ionic gelation	[32]
Curcumin	Magnetic microsphere	Lung and cervical	Inhibits cell proliferation	A549 and Hela	NI *	Dual-emulsion solvent washout	[33]
Naringenin	PLGA	Pancreatic	Inhibits cell proliferation	NI *	150.45 ± 12.45 nm	Emulsion diffusion evaporation	[34]
Naringenin	Chitosan	Lung	Inhibits cell proliferation	A549	407.47 nm	Ionic gelation	[35]
Naringenin	PHEMA	Breast	Cell cycle arrest in G1 phase and induces early apoptosis	MCF-7	53 ± 1.1 nm	Mini-emulsion polymerization	[36]
Naringenin	PCL	Lung	Cell cycle arrest in G2-M phase	A549	251.6 ± 3.22 nm	Nano-precipitation	[37]
Naringenin	Polymeric nano micelles	Lung and breast	Inhibits cell proliferation	A549 and MCF-7	239.8 ± 0.76 nm	Membrane hydration method	[38]
Quercetin	Eudragit® S100	Colon	Inhibits cell proliferation	CT26	66.8 nm	Nano-precipitation	[39]
Quercetin	Chitosan	Lung and breast	Increases cytotoxic activity	A549 and MCF-7	339.37 nm	Ionic gelation	[40]
Quercetin	PLGA	Breast and cervical	Induces apoptosis, mitochondrial damage, caspase activation, and cell cycle arrest	MCF-7 and Hela	89.8 ± 5.9 nm	Solvent evaporation	[41]
Quercetin	PLGA/TPGS	Breast	Decreases migration and invasion	MDA-MB-231	198.4 ± 7.8 nm	Nanoprecipitation	[42]
Quercetin	PLGA	Colon	Increases Cas-3 and Cas-9 expression	HT-29	237.8 ± 9.67 nm	Solvent evaporation	[43]
Quercetin	PLGA	Liver	Enhances Cyt-c/caspase and Akt/ERK1/2, AP-2β/hTERT, and p65/COX-2 signal inactivation	MHCC97H, Hep3B, HCCLM3, and Bel7402	106.7 nm	NI *	[44]
Epigallo-catechin gallate	PLGA	Prostate	Increases apoptotic cells and mitochondrial depolarization	PC-3 and 22Rv1	115 to 130 nm	Nanoprecipitation	[45]
Epigallo-catechin gallate	PLGA	Lung	Blocks NF-κB activity and suppresses genes regulated by NF-κB such as BCL2, BCL-XL, COX-2, TNF-a, cyclinD1, C-MYC, TWIST, and MMP2	A549 and H1299	175.8 ± 3.8 nm	Oil-in-water emulsion solvent evaporation	[46]
Epigallo-catechin gallate	Chitosan	Breast	Inhibits cell proliferation	MCF-7	342 nm	Ionic gelation	[47]
Thymoquinone	PLGA	Colon	Inhibits cell proliferation	C26 and Caco-2	184 nm	Single-emulsion-solvent evaporation	[48]
Thymoquinone	PLGA-PEG	Breast	Inhibits cell proliferation	MDA-MB-231 and MCF-7	6.92 ± 27.38 nm	Emulsion-solvent evaporation	[49]
Thymoquinone	PLGA	Breast	Greater muco-adhesion, intestinal permeability, and enhanced antioxidant potential and cytotoxicity	MDA-MB-231 and MCF-7	126.03–196.71 nm	Emulsion evaporation	[50]
Thymoquinone	PLGA-PEG	Lung	TF-TQ-Np-mediated p53 upregulation and activated miR-34a and miR-16 expression levels	A549	77.50 ± 6.35 nm	Nanoprecipitation	[51]
Thymoquinone	Chitosan	Ovarian	Inhibits cell proliferation	SKOV-3	250 to 350 nm	Ionic gelation	[52]
Thymoquinone	PLGA	Melanoma	Inhibits cell proliferation	A375	147.2 nm	Multiple-emulsion-solvent diffusion	[53]
Kaempferol	PEO-PPO-PEO, PLGA, PLGA-PEI, chitosan, and PAMAM	Ovarian	Inhibits cell proliferation	A2780/CP70 and OVCAR-3	160, 210, 220, 230, and 250 nm	Nanoprecipitation	[54]
Resveratrol	PEG-PLA	Colon	Induced apoptosis	CT26	119.9 nm	Solvent-evaporation method	[55]
Resveratrol	PLGA	Prostate	G1-S transition phase, externalization of phosphatidylserine, DNA nicking, loss of mitochondrial membrane potential, and reactive oxygen species generation	LNCaP	202.8 nm	Nanoprecipitation	[56]
Resveratrol	PLGA	Prostate	Inhibits cell proliferation	DU-145, PC-3, and LNCaP	150 nm	Nanoprecipitation	[57]
Resveratrol	PLGA-TPGS	Brain	Inhibits cell proliferation	C6	175.5 nm	Single-emulsion-solvent- evaporation	[58]
Resveratrol	mPEG-PLA	Melanoma	Inhibits cell proliferation	B16-F10	162.2 ± 2.9 nm	Solvent evaporation	[59]
Genistein	PLGA	Glioblastoma	Increases levels of Cyt-c, Cas-3, Cas-9, and BAX gene and Cas-3 and Cas-9 protein expression	U87MG	135.5 to 180.7 nm	Single-emulsion (o/w) solvent evaporation	[60]
Genistein	PLGA-PEG	Ovarian	Inhibits cell proliferation	SKOV-3	104.17 to 125.41 nm	Nanoprecipitation	[61]
Genistein	Chitosan	Colorectal	Inhibits cell proliferation	HT-29	788 nm	Ionic gelation	[62]
Apigenin	PLGA	Melanoma and lung	Depolarizes mitochondrial membrane potential, enhances caspase activity	B16-F10 and A549	92.18 nm	Nanoprecipitation	[63]

* NI, no information.

## Data Availability

Not applicable.

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
