# Peer review of "Recent Strategies for Cancer Therapy: Polymer Nanoparticles Carrying Medicinally Important Phytochemicals and Their Cellular Targets"

_pharmaceutics, 2023, doi:10.3390/pharmaceutics15112566_

Round 1

Reviewer 1 Report (New Reviewer)

Comments and Suggestions for Authors

The manuscript pharmaceutics-2585302 “Recent Strategies for Cancer Therapy: Polymer Nanoparticles Carrying Medicinally Important Phytochemicals and Their Cellular Targets” by Metin Yıldırım et al. reviews the production and characterization of polymeric nanoparticles loaded with phytochemical compounds that show both antitumor activity and significant antioxidant and anti-inflammatory effects, and their possible cellular targets against various cancers are presented. The review is well written, however some additions are needed. The paper will definitely be of interest to the readers of Pharmaceutics.

Questions and comments:

  1. Line 84 - Polymer particle size for antitumor drug delivery is typically 50 to 300-500 nm.  Incidentally, this is confirmed by the data in Table 1. Moreover, non-parenteral formulations can reach sizes larger than 300-500 nm.

  2. A Section describing the advantages of polymeric particles for anticancer therapy should be added. The use of polymeric particles not only increases the solubility of poorly soluble antitumor agents, but also increases their bioavailability, decreases the drug dose and frequency of administration, and reduces toxicity and side effects. Moreover, nanoparticles are able to provide targeted delivery to the tumor through both active targeting (affinity to various receptors expressed in the tumor, such as CD44, etc.) and passive targeting (EPR effect). In addition, polymeric particles can be used to program and control drug release in tumors through stimulus-sensitivity (release under the influence of pH, glutathione, redox potential, etc.).

  3. Some information on polymers that have antitumor potential and can be used to produce nanoparticles with improved antitumor activity (e.g., fucoidan, doi.org/10.3390/ijms24032615) should be added.

  4. Information on the possible cellular targets of phytochemical compounds against different types of cancer should be summarized. This is a really interesting topic declared in the abstract.

Comments on the Quality of English Language

 Minor editing of English language required

Author Response

We thank the reviewers for their insightful comments on the manuscript. Our response to their comments are given below:

Reviewer 1

Line 84 - Polymer particle size for antitumor drug delivery is typically 50 to 300-500 nm.  Incidentally, this is confirmed by the data in Table 1. Moreover, non-parenteral formulations can reach sizes larger than 300-500 nm.

Answer: We have revised the manuscript according to your comment on the approximate sizes of nanoparticles. Please see the sections highlighted in green.

A Section describing the advantages of polymeric particles for anticancer therapy should be added. The use of polymeric particles not only increases the solubility of poorly soluble antitumor agents, but also increases their bioavailability, decreases the drug dose and frequency of administration, and reduces toxicity and side effects. Moreover, nanoparticles are able to provide targeted delivery to the tumor through both active targeting (affinity to various receptors expressed in the tumor, such as CD44, etc.) and passive targeting (EPR effect). In addition, polymeric particles can be used to program and control drug release in tumors through stimulus-sensitivity (release under the influence of pH, glutathione, redox potential, etc.).

Answer: We are greatly appreciative of your constructive recommendations on our paper. A section describing the advantages of polymeric particles for anticancer therapy has been placed within the text (highlighted in green).

Some information on polymers that have antitumor potential and can be used to produce nanoparticles with improved antitumor activity (e.g., fucoidan, doi.org/10.3390/ijms24032615) should be added.

Answer: The related information has been added using the suggested reference.

Information on the possible cellular targets of phytochemical compounds against different types of cancer should be summarized. This is a really interesting topic declared in the abstract.

Answer: The information which was suggested has been added (highlighted in green) under each of the phytochemicals.

Reviewer 2 Report (New Reviewer)

Comments and Suggestions for Authors

The manuscript titled ‘Recent Strategies for Cancer Therapy: Polymer Nanoparticles Carrying Medicinally Important Phytochemicals and Their Cellular Targets’ is comprehensive and well organized. Minor revision is recommended to improve the rigor of this manuscript.

1.      Include a section describing about regulatory considerations for nanoparticles.

2.      In general, it is well written, but there are some grammars and typing/spacing errors should be corrected. The English is generally satisfactory but a native speaker should read the paper and correct some sentences

Comments on the Quality of English Language

 In general, it is well written, but there are some grammars and typing/spacing errors should be corrected. The English is generally satisfactory but a native speaker should read the paper and correct some sentences

Author Response

We thank the reviewers for their insightful comments on the manuscript. Our response to their comments are given below:

Reviewer 2

Include a section describing about regulatory considerations for nanoparticles.

Answer: There are no FDA approvals or clinical studies available in the literature for phytochemical-loaded polymer nanoparticles.

In general, it is well written, but there are some grammars and typing/spacing errors should be corrected. The English is generally satisfactory but a native speaker should read the paper and correct some sentences

Answer: The entire text was reviewed again by our native speaker, and necessary revisions were made.

Reviewer 3 Report (Previous Reviewer 2)

Comments and Suggestions for Authors

I have gone through the manuscript titled “Recent Strategies for Cancer Therapy: Polymer Nanoparticles Carrying Medicinally Important Phytochemicals and Their Cellular Targets” authored by Yıldırım et al. The manuscript emphasizes recent therapy for cancer using polymer nanoparticles containing medicinally important phytochemicals. I have following submissions:

1.    The methodology and inclusion and exclusion criteria has not been defined

2.    This manuscript does not report any clinically approved such polymer nanoparticles, if any or information related to it.

3.    The manuscript has been written so casually as most of the places it lacks support of the references. for eg. line no 48, 49 on page no.2  and many more.

4.    The whole manuscript represents just the pasting of previously published reports without any flow.

5.    Almost all the figures have been taken from previously published papers, which simply signifies the lack of novelty.

6.    There are many latest reports and findings that have not been mentioned for e.g. https://doi.org/10.3390%2Fcancers15041023

In my opinion this manuscript in its current form doesn’t meet the high standards of this journal and should not be accepted.

Comments on the Quality of English Language

 Moderate editing of English language required throughout the manuscript.

Author Response

We thank the reviewers for their insightful comments on the manuscript. Our response to their comments are given below:

Reviewer 3

I have gone through the manuscript titled “Recent Strategies for Cancer Therapy: Polymer Nanoparticles Carrying Medicinally Important Phytochemicals and Their Cellular Targets” authored by Yıldırım et al. The manuscript emphasizes recent therapy for cancer using polymer nanoparticles containing medicinally important phytochemicals. I have following submissions:

  1. The methodology and inclusion and exclusion criteria has not been defined

Answer: It was not our aim to to generate a “systemetic review,” but rather a “narrative review.” Most reviews in the biomedical sciences are narrative reviews, and are based on the experience and perspective of the authors in a particular field.

  1. This manuscript does not report any clinically approved such polymer nanoparticles, if any or information related to it.

Answer: At present, we are not aware of any nanoparticle-associated phytochemicals that have been approved by the FDA.

  1. The manuscript has been written so casually as most of the places it lacks support of the references. for eg. line no 48, 49 on page no.2  and many more.

Answer: The part of the manuscript that the Reviewer refers to is in the introduction where we did not choose to include all the relevant references, which would in turn be cited in the rest of the review. We hope the Reviewer appreciates the inclusion of 126 references as part of the review.

  1. The whole manuscript represents just the pasting of previously published reports without any flow.

Answer: We chose to refer in detail to particular references as an introduction to readers who may be new to this field, or to investigators who would like to hone in on a particular nanoparticle and its physicochemical and therapeutic properties. This approach is in contrast to reviews that lump together a large number of studies without providing significant details. The two other Reviewers have found no flaws in the flow of the narrtive.

  1. Almost all the figures have been taken from previously published papers, which simply signifies the lack of novelty.

Answer: We have included additional figures in the revised manuscript at the request of the Academic Editor. The figures relate to previous, significant studies in this field that we feel are helpful in illustrating the potential of the pharmaceutical approach we have described in the review.

  1. There are many latest reports and findings that have not been mentioned for e.g. https://doi.org/10.3390%2Fcancers15041023

Answer: The paper that the Reviewer cites focuses on the use of nanocarriers based on phytochemicals and their use with established anticancer agents, and not on the polymeric nanoparticles we have discussed in our review. We have therefore chosen to not cite this paper.

In my opinion this manuscript in its current form doesn’t meet the high standards of this journal and should not be accepted.

Answer: We believe we have responded satisfactorily to the comments of the two other Reviewers, and that we have adhered to the high standards of Pharmaceutics. We have also responded to the points raised by Reviewer 3, who, unfortunately, is determined to reject our paper.

Round 2

Reviewer 1 Report (New Reviewer)

Comments and Suggestions for Authors

The manuscript may be accepted.

Comments on the Quality of English Language

English is fine.

Author Response

Thank you for your comment on our paper. 

Reviewer 3 Report (Previous Reviewer 2)

Comments and Suggestions for Authors

The authors' responses to all of the raised questions demonstrate their obstinate unprofessional demeanor and rigidity in taking comments personally.

Answer .1  It was not our aim to generate a “systemetic review,” but rather a “narrative review.” Most reviews in the biomedical sciences are narrative reviews, and are based on the experience and perspective of the authors in a particular field.

 While narrative reviews cover the literature thoroughly, they do not provide a description of the explicit search techniques used. I stressed the importance of including inclusion and exclusion criteria because narrative reviews are usually criticised for being biased in their selection of the available literature and for not discussing how the review process was carried out.

Answer: At present, we are not aware of any nanoparticle-associated phytochemicals that have been approved by the FDA.

The authors should have confirmed this as in their previous reply authors have emphasized their writing as “narrative review”.

Answer: The part of the manuscript that the Reviewer refers to is in the introduction where we did not choose to include all the relevant references, which would in turn be cited in the rest of the review. We hope the Reviewer appreciates the inclusion of 126 references as part of the review.

Please carefully read the remarks. I said "for e.g. line no. 48, 49 on page no. 2 and many more," which implies that I'm not only stating this for the Introduction section.

Answer : We chose to refer in detail to particular references as an introduction to readers who may be new to this field, or to investigators who would like to hone in on a particular nanoparticle and its physicochemical and therapeutic properties. This approach is in contrast to reviews that lump together a large number of studies without providing significant details. The two other Reviewers have found no flaws in the flow of the narrtive

Again, the authors hastily addressed the critiques. "Flow" refers to connecting together or correlating information from several research; I did not mean that important details needed to be omitted. Authors are additionally forbidden from contrasting or contesting the opinions of one reviewer with those of other reviewers. Please be considerate of your words since we are all making individual comments to help create potentially excellent articles for the journal.

Answer: The paper that the Reviewer cites focuses on the use of nanocarriers based on phytochemicals and their use with established anticancer agents, and not on the polymeric nanoparticles we have discussed in our review. We have therefore chosen to not cite this paper.

It is again requested not to reply hastily, the suggested paper includes polymeric nanoparticles as well (section 1.13 HA-based polymeric NPs ) READ AGAIN.

Answer: We believe we have responded satisfactorily to the comments of the two other Reviewers, and that we have adhered to the high standards of Pharmaceutics. We have also responded to the points raised by Reviewer 3, who, unfortunately, is determined to reject our paper.

Authors should be asked to apologize for this behavior. The authors' responses above have demonstrated that they are willing to raise unfounded suspicions about the reviewer's integrity.  How can they write that the reviewer intends to reject their paper? They are doubting my integrity, and they ought to feel guilty for behaving in a way that is discourteous.

I continue to believe that the authors have displayed their adamant conduct. Furthermore, they haven't properly responded to any one of the queries. Consequently, this manuscript must have to be rejected.

Comments on the Quality of English Language

Moderate editing of English language required

Author Response

Reviewer 3

We thank the reviewer for reviewing the revised manuscript. We have responded to all the scientific criticism presented by the Reviewer. Below are our responses to the points raised by the Reviewer.

While narrative reviews cover the literature thoroughly, they do not provide a description of the explicit search techniques used. I stressed the importance of including inclusion and exclusion criteria because narrative reviews are usually criticised for being biased in their selection of the available literature and for not discussing how the review process was carried out.

Answer: We have merely followed the conventional approach to generating review articles. Many published reviews acknowledge that they may not have covered all the literature. We tried to describe most of the studies we could find that dealt with the therapeutic use of phytochemical nanoparticles against cancer.

The authors should have confirmed this as in their previous reply authors have emphasized their writing as “narrative review”.

Answer: We have searched the literature for clinical trials involving phytochemical-incorporating nanoparticles, and found three studies with curcumin nanoparticles. We have described these studies at the end of the section on Curcumin. We have indicated at the end of sections describing the other phytochemicals that we have not found any clinical trials on nanoparticles encapsulating these phytochemicals.

Please carefully read the remarks. I said "for e.g. line no. 48, 49 on page no. 2 and many more," which implies that I'm not only stating this for the Introduction section.

Answer: We believe that we have included a large number of studies in our review. Reviews that are too long become too unwieldy and are not read by scientists who have limited time.

Again, the authors hastily addressed the critiques. "Flow" refers to connecting together or correlating information from several research; I did not mean that important details needed to be omitted. Authors are additionally forbidden from contrasting or contesting the opinions of one reviewer with those of other reviewers. Please be considerate of your words since we are all making individual comments to help create potentially excellent articles for the journal.

Answer: Our review covers many of the studies we found in the literature, giving sufficient details on each one such that the studies can be compared by the reader. We leave it to the reader to compare the studies and utilize their significant findings in their own research.

We are not aware of rules preventing authors from referring to the comments of other reviewers while responding to a particular reviewer.

It is again requested not to reply hastily, the suggested paper includes polymeric nanoparticles as well (section 1.13 HA-based polymeric NPs ) READ AGAIN.

Answer: We have referred to the paper suggested by the reviewer in the Conclusions section, as the paper points out the potential problem of the amount of phytochemicals that can be incorporated in nanoparticles.

Authors should be asked to apologize for this behavior. The authors' responses above have demonstrated that they are willing to raise unfounded suspicions about the reviewer's integrity.  How can they write that the reviewer intends to reject their paper? They are doubting my integrity, and they ought to feel guilty for behaving in a way that is discourteous.

I continue to believe that the authors have displayed their adamant conduct. Furthermore, they haven't properly responded to any one of the queries. Consequently, this manuscript must have to be rejected.

Answer: We have had no such intentions regarding Reviewer 3’s integrity, and we are sorry that our sentences were construed this way. The review process is based on facts. We have responded to all the factual criticism, and have modified the manuscript extensively over the course of 3 rounds of review.

Round 3

Reviewer 3 Report (Previous Reviewer 2)

Comments and Suggestions for Authors

I still contend that the paper should not be published because the authors are so adamant about their own opinions and didn't take into account the suggestions.

Comments on the Quality of English Language

Moderate editing of English language required

Author Response

We have responded to a number of the suggestions of the Reviewer. Please see the Editor's suggestions and our response to them.

This manuscript is a resubmission of an earlier submission. The following is a list of the peer review reports and author responses from that submission.

Round 1

Reviewer 1 Report

Comments and Suggestions for Authors

This manuscript provided an insightful, comprehensive, and latest review about using nanoparticle-based polymers as carriers to enhance the in vitro and in vivo anticancer activity of diverse phytochemicals. This manuscript is suitable for publication in its present format.

Author Response

Reviewer 1 states: "This manuscript provided an insightful, comprehensive, and latest review about using nanoparticle-based polymers as carriers to enhance the in vitro and in vivo anticancer activity of diverse phytochemicals. This manuscript is suitable for publication in its present format."

We greatly appreciate the recognition of the importance of our review and of the suitability of its publication.

Reviewer 2 Report

Comments and Suggestions for Authors

I have gone through the manuscript titled “Recent Strategies for Cancer Therapy: Polymer Nanoparticles Carrying Medicinally Important Phytochemicals and Their Cellular Targets” by Yıldırım et al. I have following submissions

Foremost thing is a novelty of the subject. there are many comprehensive reviews already available in this subject. Additionally there IS LACK OF FLOW in the manuscript. Mere informations are pasted as such in each section, there is no flow between the lines.

Review lack many updated references

review lacks phytonanomedicine that are approved by FDA or in clinical trials for the treatment of cancer. Clinical studies are lacking in this review

Along with these many other mistakes are as follows:

Line 38-39 “The term “phytochemical” was First coined in the 1950s, but the medicinal effects of 38 phytochemicals have been known for centuries”. It’s a bold statement please provide reference

In table 1 what does NI mean?? Provide its full form.

In the caption of fig 1 ‘Phytochemical compounds loaded in polymeric NPs and their anticancer activities” where the anticancer activity is shown?? this is an irrelevant figure as the same has been depicted in table 1 with better level of understanding.

Language needs to be improvised many places for e.g. line 76 “Curcumin (Cur), a polyphenol found in ground rhizomes of Curcuma longa L., a plant”

If this paper focuses on Cancer then what is the reason for including the reports of Alzheimer’s disease (Line 136-140)

In line no 473-475 “In this review, the anticancer efficacy of natural phytochemical compounds, which  have low anticancer activity on their own due to limitations such as low bioavailability, solubility in water, and stability, was examined when they were loaded into different polymer NPs.

the highlighted line should be deleted as many of these already well established anticancer agent as such and one cannot write such a bold statement that due to low bioavailability the exhibit low anticancer activity because not every reported compound comes in this category

In my opinion manuscript cannot be accepted for publication in this journal.

Comments on the Quality of English Language

I have gone through the manuscript titled “Recent Strategies for Cancer Therapy: Polymer Nanoparticles Carrying Medicinally Important Phytochemicals and Their Cellular Targets” by Yıldırım et al. I have following submissions

Foremost thing is a novelty of the subject. there are many comprehensive reviews already available in this subject. Additionally there IS LACK OF FLOW in the manuscript. Mere informations are pasted as such in each section, there is no flow between the lines.

Review lack many updated references

review lacks phytonanomedicine that are approved by FDA or in clinical trials for the treatment of cancer. Clinical studies are lacking in this review

Along with these many other mistakes are as follows:

Line 38-39 “The term “phytochemical” was First coined in the 1950s, but the medicinal effects of 38 phytochemicals have been known for centuries”. It’s a bold statement please provide reference

In table 1 what does NI mean?? Provide its full form.

In the caption of fig 1 ‘Phytochemical compounds loaded in polymeric NPs and their anticancer activities” where the anticancer activity is shown?? this is an irrelevant figure as the same has been depicted in table 1 with better level of understanding.

Language needs to be improvised many places for e.g. line 76 “Curcumin (Cur), a polyphenol found in ground rhizomes of Curcuma longa L., a plant”

If this paper focuses on Cancer then what is the reason for including the reports of Alzheimer’s disease (Line 136-140)

In line no 473-475 “In this review, the anticancer efficacy of natural phytochemical compounds, which  have low anticancer activity on their own due to limitations such as low bioavailability, solubility in water, and stability, was examined when they were loaded into different polymer NPs.

the highlighted line should be deleted as many of these already well established anticancer agent as such and one cannot write such a bold statement that due to low bioavailability the exhibit low anticancer activity because not every reported compound comes in this category

In my opinion manuscript cannot be accepted for publication in this journal.

Author Response

Review lacks many updated references.

Answer: A careful examination of the references will show that the studies we have mentioned in the review are quite new.

Review lacks phytonanomedicine that are approved by FDA or in clinical trials for the treatment of cancer. Clinical studies are lacking in this review

Answer: There is no FDA approval or clinical study available in the literature for phytochemical-loaded polymer nanoparticles.

Along with these many other mistakes are as follows:

Line 38-39 “The term “phytochemical” was First coined in the 1950s, but the medicinal effects of 38 phytochemicals have been known for centuries”. It’s a bold statement please provide reference

Answer: The reference has been added.

In table 1 what does NI mean?? Provide its full form.

Answer: The abbreviation "NI" has been defined at the bottom of the table, and the "NI" in the table is followed by an asterisk.

In the caption of fig 1 ‘Phytochemical compounds loaded in polymeric NPs and their anticancer activities” where the anticancer activity is shown?? this is an irrelevant figure as the same has been depicted in table 1 with better level of understanding.

Answer: It is written next to the cancer on which the active substances have an effect. We believe this figure is a very useful summary of the effects of the different phytochemical reviewed in the manuscript.

Language needs to be improvised many places for e.g. line 76 “Curcumin (Cur), a polyphenol found in ground rhizomes of Curcuma longa L., a plant”

Answer: This review was co-written and edited by a native speaker.

If this paper focuses on Cancer then what is the reason for including the reports of Alzheimer’s disease (Line 136-140)

Answer: Upon the recommendation of the reviewer, this part has been deleted.

In line no 473-475 “In this review, the anticancer efficacy of natural phytochemical compounds, which  have low anticancer activity on their own due to limitations such as low bioavailability, solubility in water, and stability, was examined when they were loaded into different polymer NPs.

the highlighted line should be deleted as many of these already well established anticancer agent as such and one cannot write such a bold statement that due to low bioavailability the exhibit low anticancer activity because not every reported compound comes in this category

Answer: As requested, this part has been deleted.

Reviewer 3 Report

Comments and Suggestions for Authors

This manuscript aims at describing the preparation and characterization of natural phytochemical-loaded polymer nanoparticles formulations with important biological activity such as antioxidant and anticancer effects, as well as their possible cellular targets against various types of cancers. I would like to recommend the publication of this manuscript in Pharmaceutics subjected to major revision.

 1.     In abstract, the authors claim that in the review article they would describe the preparation and characterization of phytochemical-loaded polymer nanoparticles formulations. However, as far as I understand, their description were mainly focused on the bioactivity and possible cellular targets of the nanoparticles of interest, but much less description and evaluation were made to the preparation and characterization, in particle the latter, of the nanoparticles. I suggest that the authors should add more description and critical evaluation on the preparation and characterization of different phytochemical-loaded polymer nanoparticles.

2.     As far as I understand, the structures (components) and sizes of nanoparticle carries largely determine the effectiveness of loaded phytochemicals. However, the authors did not provide detailed evaluation and comparison for the effects of these two parameters of nanoparticles on biological activity of loaded phytochemicals.  

3.     I noticed that according to the authors’ description, the presence of folic acid increased the particle size of Chi-EGCD NPs from ca. 164 nm to 342 nm (lines 251 – 252), in contrast folic acid decoration only increased slightly the particle size of Gen-PLGA-PEG NPs (104 – 125 nm, lines 438 – 439). The authors should interpret what caused this large difference, and how this difference influence the bioactivity of loaded natural phytochemicals.

4. A minor comment: lines 82 – 84, the sentence starting with “Cur” is not clear, it should be read as that “Cur exhibited anticancer activity by involving in regulation of several cellular pathways ....”.    

Author Response

  1. In abstract, the authors claim that in the review article they would describe the preparation and characterization of phytochemical-loaded polymer nanoparticles formulations. However, as far as I understand, their description were mainly focused on the bioactivity and possible cellular targets of the nanoparticles of interest, but much less description and evaluation were made to the preparation and characterization, in particle the latter, of the nanoparticles. I suggest that the authors should add more description and critical evaluation on the preparation and characterization of different phytochemical-loaded polymer nanoparticles.

Answer: More descriptions and critical evaluation on the preparation and characterization of different phytochemical-loaded polymer nanoparticles are added in the revised text

  1. As far as I understand, the structures (components) and sizes of nanoparticle carries largely determine the effectiveness of loaded phytochemicals. However, the authors did not provide detailed evaluation and comparison for the effects of these two parameters of nanoparticles on biological activity of loaded phytochemicals.  

Answer: The anticancer activity of drug-loaded nanoparticles depends on many physical and biological factors. Therefore we have refrained from make direct comments on these factors.

  1. I noticed that according to the authors’ description, the presence of folic acid increased the particle size of Chi-EGCD NPs from ca. 164 nm to 342 nm (lines 251 – 252), in contrast folic acid decoration only increased slightly the particle size of Gen-PLGA-PEG NPs (104 – 125 nm, lines 438 – 439). The authors should interpret what caused this large difference, and how this difference influence the bioactivity of loaded natural phytochemicals.

Answer: The polymer structures used in the 2 studies are different from each other. It is normal for them to differ in such studies.

  1. A minor comment: lines 82 – 84, the sentence starting with “Cur” is not clear, it should be read as that “Cur exhibited anticancer activity by involving in regulation of several cellular pathways

Answer: The sentence has been edited.